# Dynamic Finite Element Model Updating Based on Correlated Mode Auto-Pairing and Adaptive Evolution Screening

**Huajin Shao** [1] , **Yanfei Zuo** [1,*] **and Zhinong Jiang** [2]

1   Key Laboratory of Engine Health Monitoring-Control and Networking of Ministry of Education, Beijing University of Chemical Technology, Beijing 100029, China; 2017400132@mail.buct.edu.cn
2   Key Laboratory of High-End Mechanical Equipment Health Monitoring and Self-Recovery, Beijing University of Chemical Technology, Beijing 100029, China; jiangzn@mail.buct.edu.cn
*   Correspondence: zuoyf@mail.buct.edu.cn; Tel.: +86-132-4190-4663

**Abstract:** A method for dynamic finite element (FE) model updating based on correlated mode auto-pairing and adaptive evolution screening (CMPES) is proposed to overcome difficulties in pairing inaccurate analytical modal data and incomplete experimental modal data. In each generation, the correlated mode pairings (CMPs) are determined by modal assurance criterion (MAC) values and the symbiotic natural frequency errors, according to an auto-pairing strategy. The objective function values constructed by correlated and penalized subitems are calculated to screen the better individuals. Then, both the updating parameters and the CMPs can be adjusted adaptively to simultaneously approach the ideal results during the iteration of population evolution screening. Three examples (a thin plate with small holes, an F-shaped structure, and an intermediate case with multi-layer thin-walled complex structure) were presented to validate the accuracy, effectiveness, and engineering application potential of the proposed method.

**Keywords:** finite element; dynamic model updating; correlation pairing; evolution screening; modal analysis

## 1. Introduction

The dynamic analyses of complex structures provided by finite element (FE) models have become increasingly important, especially for complex structures working in adverse conditions and costing a lot for experimentation [1–3]. The discrepancies between initial FE analyses and experimental results inevitably exist due to some factors such as structural simplification, inaccurate material parameters, unreasonable boundary conditions, and unknown damping characteristics [4,5]. Therefore, the dynamic FE model updating is a common practice to decrease these discrepancies [6–9]. On the other hand, the data obtained from experimental models are often incomplete due to many limitations, including the limited number, position or direction of measuring points, limitation of the measurable frequency range, and the increasing modal density with frequency increasing [10,11]. Hence, it is very difficult to pair the correlated modes between inaccurate analytical modal data and incomplete experimental modal data purely based on frequencies or/and modal assurance criterion (MAC) thresholds [12–15]. Generally, the following situations are encountered:

(1)   The correlation between analytical and experimental modes can be uniquely established by frequencies or/and MAC thresholds, denoted as case 1↔1 in this paper.

(2)   Similarly, there is a one-to-many or many-to-one correlation between analytical and experimental modes established by frequencies or/and MAC thresholds. This type of situation is denoted as case 1↔n.

(3)   Compared with measuring data, one or several analytical modes are missing. Alternatively, one or several experimental modes are missing compared with numerical results. This type of situation is denoted as case 0↔n.

Obviously, for case 1↔1, the correlated mode pairings (CMPs) can be determined unquestionably. Since the objective function values calculated by the analytical and experimental modal data residuals can be obtained directly, routine updating methods [5,10] work well. However, case 1↔n and case 0↔n are ignored and occur commonly in dynamic model updating of the complex structures, which may cause significant discrepancies in certain circumstances because some valuable analytical or experimental modal data are not fully exploited.

Improved methods by specifying correlated modes based on pre-analysis are employed to overcome the above shortages. Case 1↔n could be transformed into the ideal case 1↔1 by specifying the one-to-one CMPs manually based on appropriate frequencies or/and MAC thresholds after pre-analysis. For example, three CMPs were specified to transform case 1↔n into case 1↔1 in the beam model updating of the GARTEUR structure by Thonon and Golinval [16]. Similar improved methods were adopted by many researchers, such as the plate model updating of the GARTEUR structure by Bohle and Fritzen [17], the aero-engine casing model updating by Zang et al. [18,19] and Zhai et al. [20], the Podgorica footbridge model updating by Zivanovic [21], and the Guadalquivir bridge model updating by Tran-Ngoc [22].

Modak proposed a model updating method by using uncorrelated modes (MUUM) [23]. The correlated and uncorrelated modes were determined after pre-analysis. The objective function was innovatively defined as the quadratic sum of the relative errors in the correlated modes plus the geometric mean for the relative errors in the uncorrelated modes. Thus, uncorrelated modes of experimental data are used to simplify the specifying of CMPs and avoid manual mistakes when unreasonable updating parameter values are assigned in pre-analysis. Fei et al. developed a hierarchical model updating strategy [24] and parametric modeling-based model updating strategy [25] concerning uncorrelated modes.

All the methods mentioned above, without case 0↔n being considered, may lead to inaccurate or insufficient dynamic model updating results because some modes, especially those in experimental modal data, are ignored. In addition, constant CMPs obtained by pre-analysis may not work when the correlated or uncorrelated modes (natural frequencies or mode shapes) change dramatically over iterations, especially for complex structures with a wide range of uncertain parameters, complicated boundary conditions, and significant structural simplification. A more general and accurate updating method is needed.

According to the one-to-one symbiotic relationship between natural frequencies and mode shapes [26], it is feasible to pair and screen out the CMPs before each iteration without manual pre-analysis to exploit analytical and experimental modal data fully. Moreover, since the model updating, in essence, is the dynamic inverse problem with ill-posed multiple solutions, the optimization algorithm should have the ability to search for the global optimum solution [27,28]. The evolutionary algorithm, including genetic algorithm [29], evolutionary programming [30], and evolutionary strategy [31], has two central characters: population search strategy and information exchange among individuals in the population [32–34]. These two characters can solve the problems of the global optimal solution searching and correlated mode screening before each iteration in dynamic model updating.

The correlated mode auto-pairing and adaptive evolution screening (CMPES) method is proposed to solve the mentioned problems. A correlated mode auto-pairing strategy is proposed to determine the one-to-one CMPs adaptively before each iteration without pre-analysis and is suitable for all the 1↔1, 1↔n and 0↔n cases in dynamic model updating. The objective function constructed by correlated and penalized subitems of the initial generation can be determined based on this strategy. Then, the population evolutionary screening mechanism is employed to screen the CMPs further and search the global minimum. Meanwhile, the adaptive switch from penalized to correlated subitem during iteration can ensure that all the potential of analytical or experimental modes can be fully exploited.

The outline of this paper is as follows. In Section 2, the correlated mode pairing and screening mechanism and an objective function of optimization problem stemming from dynamic model updating are introduced briefly. Then, the CMPES method is proposed in Section 3. In Section 4, a thin plate with small holes, an F-shaped structure, and an intermediate case (IMC) of a gas turbine were presented to validate the accuracy, effectiveness, and engineering application potential of the proposed method. Section 5 is the conclusion.

## 2. Basic Dynamic Model Updating by Pairing Correlated Modes

The equation of free motion for a complex undamped structure, modeled by FE, is given by

$$\mathbf{M}\ddot{\mathbf{x}}(t) + \mathbf{K}\mathbf{x}(t) = 0 \tag{1}$$

where **M** and **K** are, respectively, the mass and stiffness matrices. $\mathbf{x}(t)$ and $\ddot{\mathbf{x}}(t)$ are displacement and acceleration vectors, respectively. The generalized eigenvalue problem is

$$(\mathbf{K} - \lambda\mathbf{M})\boldsymbol{\varphi} = 0 \tag{2}$$

where $\lambda$ and $\boldsymbol{\varphi}$ are the eigenvalues and eigenvectors, respectively. The solution to the eigenvalue problem of Equation (2) gives the natural frequencies and mode shapes. Then the frequency relative error (FRE) $FRE_{ik}$ for the pair of the $i^{\text{th}}$ analytical mode natural frequency $f_{A,i}$ and the $k^{\text{th}}$ experimental mode natural frequency $f_{E,k}$ can be obtained by

$$FRE_{ik} = \frac{f_{A,i} - f_{E,k}}{f_{E,k}} \times 100\% \tag{3}$$

where "$A$" and "$E$" as subscripts represent analytical mode and experimental mode, respectively.

The corresponding $MAC_{ik}$ value for the pair of the $i^{\text{th}}$ analytical mode shape $\boldsymbol{\varphi}_{A,i}$ and the $k^{\text{th}}$ experimental mode shape $\boldsymbol{\varphi}_{E,k}$ can be calculated by

$$MAC_{ik} = \frac{\left(\boldsymbol{\varphi}_{A,i}^T \boldsymbol{\varphi}_{E,k}\right)^2}{\left(\boldsymbol{\varphi}_{A,i}^T \boldsymbol{\varphi}_{A,i}\right)\left(\boldsymbol{\varphi}_{E,k}^T \boldsymbol{\varphi}_{E,k}\right)} \tag{4}$$

The $MAC_{ik}$ close to 1 indicates a good correlation between $\boldsymbol{\varphi}_{A,i}$ and $\boldsymbol{\varphi}_{E,k}$, while when it is close to 0 it indicates a poor correlation between them.

The problem of FE model updating in structural dynamics can be solved as a constrained optimization problem:

$$\begin{aligned} &\min J(\boldsymbol{\theta}) \\ subject \quad to \quad & \\ &a_i \leq \theta_i \leq b_i \quad (i = 1, 2, \cdots, N_\theta) \end{aligned} \tag{5}$$

where $\boldsymbol{\theta} = \begin{bmatrix} \theta_1 & \theta_2 & \cdots & \theta_{N_\theta} \end{bmatrix}^T$ is the vector of updating parameters, $N_\theta$ is the number of updating parameters, and $a_i$ and $b_i$ are the lower and upper boundary of the $i^{\text{th}}$ updating parameter $\theta_i$. The objective function $J(\boldsymbol{\theta})$ often used can be expressed generally as

$$\begin{aligned} J(\boldsymbol{\theta}) &= \sum_{k=1}^{N} \left[ w_{f,k}\left(\frac{f_{A,k'}(\boldsymbol{\theta}) - f_{E,k}}{f_{E,k}}\right)^2 + w_{\varphi,k}(1 - MAC_{k'k}(\boldsymbol{\theta}))^2 \right] \\ &= \sum_{k=1}^{N} \left[ w_{f,k}(FRE_{k'k}(\boldsymbol{\theta}))^2 + w_{\varphi,k}(1 - MAC_{k'k}(\boldsymbol{\theta}))^2 \right] \\ &= \sum_{k=1}^{N} J_{k'k}(\boldsymbol{\theta}) \end{aligned} \tag{6}$$

where $N$ is the number of correlated modes. $w_{f,k}$ and $w_{\varphi,k}$ are the weighting factors for the $k^{\text{th}}$ experimental natural frequency and the MAC value for the $k^{\text{th}}$ experimental mode

shape, respectively. $k'$ is the analytical mode number correlated with the $k^{th}$ experimental mode. $J_{k'k}(\theta)$ presents the subitem of the objective function $J(\theta)$ for the pair of the $k'^{th}$ analytical mode and the $k^{th}$ experimental mode. For convenience, the updating parameters and corresponding parentheses, i.e., $(\theta)$, are omitted in the following paragraph.

Thus, the primary prerequisite for dynamic model updating is to screen out the pair of the $k'^{th}$ analytical mode and the $k^{th}$ experimental mode in Equation (6). The situations described in Section 1 will be encountered when calculating the value of subitem $J_{k'k}$.

When case 1↔1 occurs only, each subitem value $J_{k'k}$ can be calculated uniquely because the one-to-one CMPs can be determined based on the given frequencies or/and MAC thresholds.

When case 1↔n occurs, the set of the one-to-one CMPs can be determined after pre-analysis, which can be expressed as

$$\mathbf{CMP} = \{ \ (A1', E1) \quad (A2', E2) \quad \cdots \quad (Ak', Ek) \quad \cdots \quad (AN'_E, EN_E) \ \} \tag{7}$$

where $(Ak', Ek)$ is the CMP between the $k'^{th}$ analytical mode and the $k^{th}$ experimental mode manual specified based on the given frequencies or/and MAC thresholds. $N_E$ is the number of experimental modes. The objective function value $J^{(r)}$ for the $r^{th}$ updating iteration can be calculated by

$$J^{(r)} = J_{1'1}^{(r)} + J_{2'2}^{(r)} + \cdots + J_{k'k}^{(r)} + \cdots + J_{N'_E N_E}^{(r)} \tag{8}$$

The constant CMPs determined by pre-analysis cannot be adjusted adaptively, even though a better correlation between the $k''^{th}$ ($k'' \neq k'$) analytical mode and the $k^{th}$ experimental mode occurs in the $r^{th}$ iteration. A better CMP for the $k^{th}$ experimental mode should be $(Ak'', Ek)$ in this circumstance. In addition, the updated results are usually susceptible to the specified CMPs. In other words, the differences among the updated models obtained by different specified CMPs may be noticeable, which will be discussed in the examples in Section 4.1.2.

When case 0↔n occurs, all the unpaired analytical or experimental modes will be discarded directly in routine methods without modal data fully exploited. For example, none of the analytical modes correlate well with the $k^{th}$ experimental mode in pre-analysis. Even though the $k'^{th}$ analytical mode correlates well with the $k^{th}$ experimental mode in the $r^{th}$ iteration, the CMP $(Ak', Ek)$ cannot be considered. Then, the situation illustrated in Section 4.2 cannot be updated well by routine methods.

## 3. Correlated Mode Auto-Pairing and Adaptive Evolution Screening Method

The CMPES method is proposed to tackle the mentioned problem of correlation pairing in the dynamic FE model updating. The correlated mode auto-pairing strategy is proposed to determine the one-to-one CMPs adaptively, and the population evolution screening mechanism is employed to screen the CMPs further and search the global minimum.

### 3.1. Correlated Mode Auto-Pairing Strategy

According to Equations (3) and (4), the matrix **FRE** and **MAC** can be expressed as

$$\mathbf{FRE} = \begin{bmatrix} FRE_{11} & FRE_{12} & \cdots & FRE_{1N_E} \\ FRE_{21} & FRE_{22} & \cdots & FRE_{2N_E} \\ \vdots & \vdots & \ddots & \vdots \\ FRE_{N_A 1} & FRE_{N_A 2} & \cdots & FRE_{N_A N_E} \end{bmatrix} \tag{9}$$

$$\mathbf{MAC} = \begin{bmatrix} MAC_{11} & MAC_{12} & \cdots & MAC_{1N_E} \\ MAC_{21} & MAC_{22} & \cdots & MAC_{2N_E} \\ \vdots & \vdots & \ddots & \vdots \\ MAC_{N_A 1} & MAC_{N_A 2} & \cdots & MAC_{N_A N_E} \end{bmatrix} \tag{10}$$

where $N_A$ and $N_E$ are the number of analytical and experiential modes, respectively.

As mentioned above, when natural frequencies or mode shapes change dramatically over iterations for complex structures, the constant $k'$ in Equation (6) determined by pre-analysis cannot be adjusted adaptively even though a better correlation occurs. In order to auto-pair and adaptively screen out the CMPs based on the one-to-one symbiotic relationship, correlated and penalized subitems are introduced as follows:

For case 1↔1 and case 1↔n, the $k'^{th}$ analytical mode is auto-paired with the $k^{th}$ experimental mode, where $MAC_{k'k}$ is maximum and greater than a given MAC threshold in the $k^{th}$ column in **MAC**. Then, the correlated subitem of the objective function $J_{k'k}$ in Equation (6) can be calculated by

$$J_{k'k} = w_{f,k} \cdot \left( \kappa_{f,k} \cdot FRE_{k'k} \right)^2 + w_{\varphi,k} \left[ \kappa_{\varphi,k} \cdot (1 - MAC_{k'k}) \right]^2 \tag{11}$$

where $w_{f,k}$ and $w_{\varphi,k}$ are taken as unity to give equal weight for each mode. The penalized function $\kappa$ can be selected or specified based on concrete issues. As given in Equation (12), a sigmoid function flipped horizontally was taken in this paper to accelerate the convergence. As shown in Figure 1, when $MAC_{k'k}$ is close to 0, the objective function value is multiplied with a bigger $\kappa$ (next to 3 here) to play a penalized role. The decreasing rate is much more significant in the middle region to urge $MAC_{k'k}$ forward to a larger value. Until $MAC_{k'k}$ is close to 1, $\kappa$ approaches 1 to remain unpenalized.

$$\kappa_{f,k} = \kappa_{\varphi,k} = 1 + \frac{2}{1 + \exp[10 - 20(1 - MAC_{k'k})]} \tag{12}$$

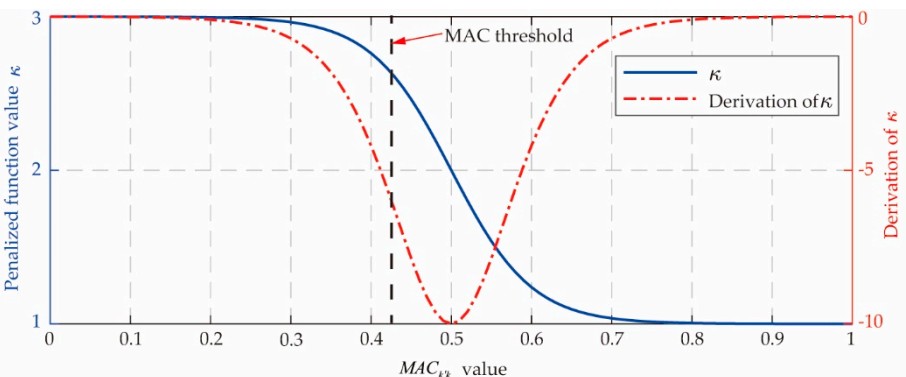

**Figure 1.** Penalized function value and its derivation used in this paper.

For case 0↔n, none of the $N_A$ analytical modes can be correlated with the $k^{th}$ experimental mode by the given MAC threshold. Constant penalized values $\overline{FRE}$ and $\overline{MAC}$, 100% and 0, respectively, in this paper, are set to guarantee equal numbers of the subitems in the objective function. Then, the penalized subitem can be calculated by

$$\overline{J_{k'k}} = w_{f,k} \cdot \left( \kappa_{f,k} \cdot \overline{FRE} \right)^2 + w_{\varphi,k} \left[ \kappa_{\varphi,k} \cdot (1 - \overline{MAC}) \right]^2 \tag{13}$$

According to Equations (11) and (13), all the subitems of the objective function can be determined by correlated mode auto-pairing strategy. In particular, when the updating parameters approach the ideal values during iteration, some analytical modes may be correlated with the experimental modes corresponding to penalized subitems. That is to say, case 0↔n may transform into case 1↔1 or case 1↔n, then the subitems calculated by Equation (13) can switch to correlated ones calculated by Equation (11). All the potential of analytical or experimental modes can be fully exploited in the updating procedure.

### 3.2. Population Evolution Screening Mechanism Provided by an Evolutionary Algorithm

The objective function value of each individual can be determined by the correlated mode auto-pairing strategy. In addition, a global optimization algorithm is needed to search the minimum value of the objective function. As mentioned in Section 1, the evolutionary algorithm has two central characters: population search strategy and information exchange among individuals in the population. Thus, the population evolution screening mechanism is particularly suitable for the CMPs adaptively screening before each iteration and the global optimal solution searching.

In evolutionary iteration, the updating parameters $\theta^{(r)}$ can be randomly generated

$$\theta^{(r)} = \begin{bmatrix} \theta^{(r,1)} & \theta^{(r,2)} & \cdots & \theta^{(r,s)} \end{bmatrix} \tag{14}$$

where "$r$" and "$s$" as superscripts represent generation number and individual number, respectively. Then, the analytical modal analyses are performed based on the FE model, and natural frequencies $\mathbf{f}_A$ and mode shapes $\boldsymbol{\varphi}_A$ for the $r^{\text{th}}$ generation can be obtained:

$$\begin{cases} \mathbf{f}_A^{(r)} = \begin{bmatrix} \mathbf{f}_A^{(r,1)} & \mathbf{f}_A^{(r,2)} & \cdots & \mathbf{f}_A^{(r,s)} \end{bmatrix} \\ \boldsymbol{\varphi}_A^{(r)} = \begin{bmatrix} \boldsymbol{\varphi}_A^{(r,1)} & \boldsymbol{\varphi}_A^{(r,2)} & \cdots & \boldsymbol{\varphi}_A^{(r,s)} \end{bmatrix} \end{cases} \tag{15}$$

Based on experimental natural frequencies $\mathbf{f}_E$ and mode shapes $\boldsymbol{\varphi}_E$, the FRE and MAC matrix for the $r^{\text{th}}$ generation can be calculated to provide the base for the following auto-pairing process.

$$\begin{cases} \mathbf{FRE}^{(r)} = \begin{bmatrix} \mathbf{FRE}^{(r,1)} & \mathbf{FRE}^{(r,2)} & \cdots & \mathbf{FRE}^{(r,s)} \end{bmatrix} \\ \mathbf{MAC}^{(r)} = \begin{bmatrix} \mathbf{MAC}^{(r,1)} & \mathbf{MAC}^{(r,2)} & \cdots & \mathbf{MAC}^{(r,s)} \end{bmatrix} \end{cases} \tag{16}$$

According to the correlated mode auto-pairing strategy, the one-to-one CMPs for the $r^{\text{th}}$ generation can be determined by the given MAC threshold in Figure 1, taken as 0.5 in this paper.

$$\mathbf{CMP}^{(r)} = \begin{Bmatrix} \mathbf{CMP}^{(r,1)} & \mathbf{CMP}^{(r,2)} & \cdots & \mathbf{CMP}^{(r,s)} \end{Bmatrix} \tag{17}$$

According to Equation (11) to Equation (13), the objective function value for the $i^{\text{th}}$ individual in the $r^{\text{th}}$ generation is given by

$$J^{(r,i)} = \sum_{k=1}^{N_E} J_{k'k}^{(r,i)} \tag{18}$$

The objective function values of all the individuals in the $r^{\text{th}}$ generation can be expressed as

$$\mathbf{J}^{(r)} = \begin{bmatrix} J^{(r,1)} & J^{(r,2)} & \cdots & J^{(r,s)} \end{bmatrix} \tag{19}$$

Based on the objective function values $\mathbf{J}^{(r)}$, rank and evaluation of all the individuals in the $r^{\text{th}}$ generation can be performed. Then $\theta^{(r+1)}$, $\mathbf{CMP}^{(r+1)}$, and $\mathbf{J}^{(r+1)}$ of the next generation can be generated by exchanging individual genes with evolutionary operators such as crossover, mutation, and selection.

$$\begin{cases} \theta^{(r+1)} = \begin{bmatrix} \theta^{(r+1,1)} & \theta^{(r+1,2)} & \cdots & \theta^{(r+1,s)} \end{bmatrix} \\ \mathbf{CMP}^{(r+1)} = \begin{Bmatrix} \mathbf{CMP}^{(r+1,1)} & \mathbf{CMP}^{(r+1,2)} & \cdots & \mathbf{CMP}^{(r+1,s)} \end{Bmatrix} \\ \mathbf{J}^{(r+1)} = \begin{bmatrix} J^{(r+1,1)} & J^{(r+1,2)} & \cdots & J^{(r+1,s)} \end{bmatrix} \end{cases} \tag{20}$$

Combined with the correlated mode auto-pairing strategy, the updating parameters $\theta$ and CMPs will simultaneously evolve toward the ideal values due to the schema theorem

and building block hypothesis [35]. The evolution process will stop until the evolutionary algorithm meets the termination condition such as the maximum number of generations has been reached or the best individual in the population has not been improved for several consecutive generations. The best individual with the minimum objective function value in the last generation is selected as the final updated result.

The procedure of the CMPES method is illustrated in Figure 2. The one-to-one $\mathbf{CMP}^{(r)}$ are determined adaptively based on the given MAC threshold. $\mathbf{J}^{(r)}$ constructed by correlated and penalized subitems are calculated based on the correlated mode auto-pairing strategy. Then, the population evolutionary screening mechanism is employed to screen the CMPs further and search the global minimum by urging θ and the CMPs towards the ideal results. Combined with the auto-pairing strategy, the adaptive switch from penalized to correlated subitem during iteration can ensure that all the potential of analytical or experimental modes can be fully exploited. The updated results will be obtained when the evolutionary algorithm meets the termination condition.

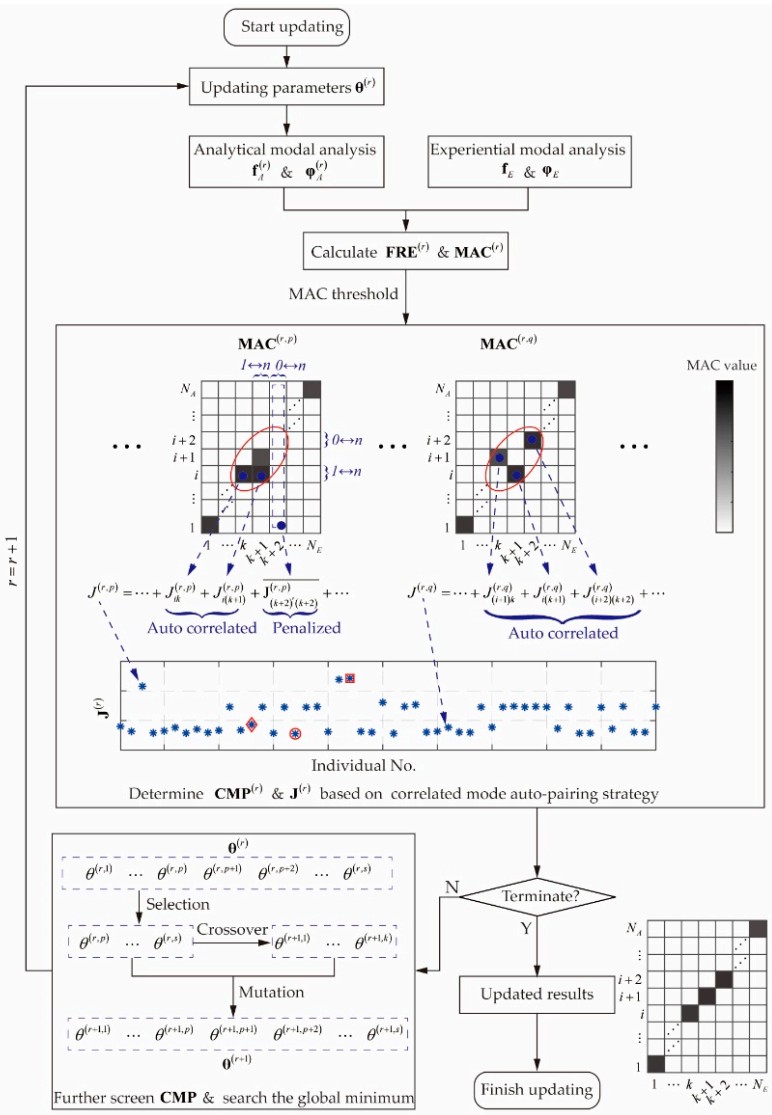

**Figure 2.** Procedure of the proposed CMPES method.

### 3.3. Dynamic Model Updating Evaluation Criteria

The root mean square errors (RMSEs) and the coefficient of variation are used to evaluate the updating results:

$$RMSE_{FRE} = \sqrt{\frac{1}{N_E}\sum_{k=1}^{N_E}(FRE_{k'k})^2} \tag{21}$$

$$RMSE_{MAC} = \sqrt{\frac{1}{N_E}\sum_{k=1}^{N_E}(1 - MAC_{k'k})^2} \tag{22}$$

$$CV = \frac{\sqrt{\frac{1}{N_E-1}\sum_{k=1}^{N_E}\left(MAC_{k'k} - \frac{1}{N_E}\sum_{k=1}^{N_E}MAC_{k'k}\right)^2}}{\frac{1}{N_E}\sum_{k=1}^{N_E}MAC_{k'k}} \tag{23}$$

where $RMSE_{FRE}$ and $RMSE_{MAC}$ are the RMSE of FRE values and the differences between MAC values and 1, respectively. $CV$ is the coefficient of variation of MAC values. A lower $RMSE_{FRE}$ or $RMSE_{MAC}$ indicates a minor discrepancy between FE and experimental results. The smaller $CV$ is, the smaller the dispersion degree will be.

## 4. Dynamic Model Updating Examples

### 4.1. Proof Examples: Dynamic Model Updating of a Thin Plate

A thin plate with non-uniform thickness and small holes is investigated to test the proposed CMPES method, as shown in Figure 3a. The average thickness, length, and width are 8, 530, and 250 mm. The diameter of all the holes is 24 mm, and the location of these holes can be determined as follows: $x_1 = 84$ mm, $x_2 = 74$ mm, $x_3 = 156$ mm, $x_4 = 129$ mm, $y_1 = y_2 = 68$ mm, $y_3 = y_4 = 122$ mm, $y_5 = y_6 = 141$ mm.

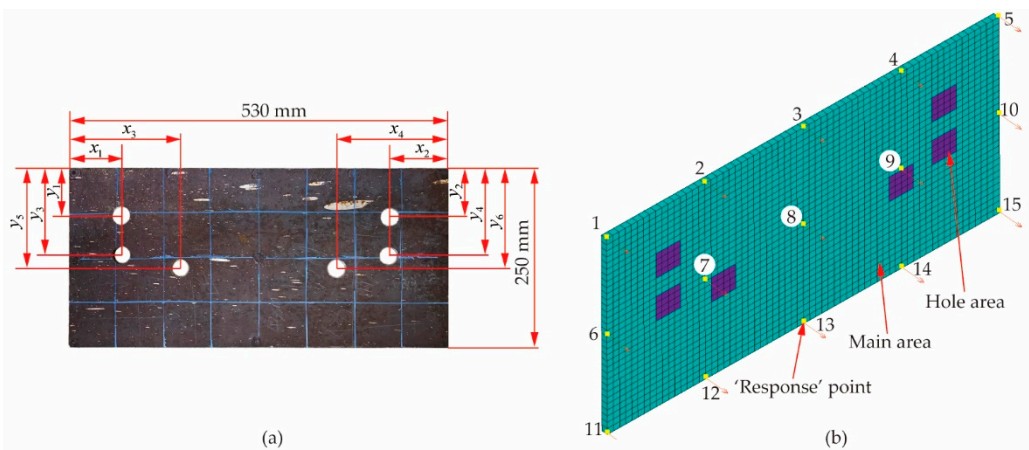

**Figure 3.** Real structure and initial FE model of the thin plate. (**a**) Dimension. (**b**) Initial FE model.

The initial FE model, ignoring the irregular geometrical characteristics, was produced by 2048 solid elements, as illustrated in Figure 3b. The equivalent areas with different material parameters were employed to present the hole areas distinguished by colors. The density, modulus of elasticity, and Poisson's ratio of steel material are 7850 kg/m³, 210 GPa, and 0.3, respectively, and the boundary condition is free–free.

Non-dimensional ratios of material parameters to the corresponding steel values were adopted as updating parameters, as listed in Table 1. The initial FE model was updated based on the simulated "experimental" and impact modal test results by the MUUM [23] and CMPES method to validate the accuracy and effectiveness of the proposed method.

**Table 1.** Material parameters for the initial FE and simulated "experimental" model of the thin plate.

| Model Type | Area Type | Non-Dimensional Ratio of Density | Non-Dimensional Ratio of Modulus of Elasticity | Non-Dimensional Ratio of Poisson's Ratio |
|---|---|---|---|---|
| Initial FE | Hole | 0.5 | 1.2 | 0.75 |
| | Main | 1.0 | 1.0 | 1.0 |
| 'Experimental' | Hole | 0.89 | 1.05 | 0.97 |
| | Main | 0.64 | 0.88 | 1.07 |

4.1.1. Updating Based on the Simulated "Experimental" Data

The "experimental" model, generating the simulated "experimental" modal data, was obtained by introducing certainly known discrepancies in the non-dimensional ratios of material parameters based on the initial FE model, as used in Reference [23], shown in Table 1.

The first ten (ignore the rigid modes) natural frequencies and mode shapes corresponding to the fifteen "response" points (see Figure 3b) were determined. Table 2 shows that all the MAC values approximated 1. However, the differences regarding natural frequencies between the initial analytical and simulated "experimental" results were remarkable, and the FRE values were about 40%. Furthermore, the $RMSE_{FRE}$, $RMSE_{MAC}$, and $CV$ values were 41.23%, 1.26%, and 1.07%, respectively. To update the initial FE model, the lower and upper boundaries of the updating parameters are given in Table 3.

**Table 2.** Comparisons between the initial analytical and simulated "experimental" modal results.

| Mode No. | "Experimental"Frequency (Hz) | Initial Analytical Modal Analysis | | | |
|---|---|---|---|---|---|
| | | Frequency (Hz) | FRE (%) | MAC | 1-MAC |
| 1 | 161.13 | 227.18 | 41.00 | 1.00 | 0.00 |
| 2 | 209.56 | 303.10 | 44.64 | 1.00 | 0.00 |
| 3 | 445.72 | 624.76 | 40.17 | 1.00 | 0.00 |
| 4 | 455.18 | 658.21 | 44.61 | 1.00 | 0.00 |
| 5 | 732.77 | 1016.49 | 38.72 | 0.98 | 0.02 |
| 6 | 780.83 | 1124.19 | 43.97 | 1.00 | 0.00 |
| 7 | 855.55 | 1175.48 | 37.40 | 0.99 | 0.01 |
| 8 | 892.67 | 1222.67 | 36.97 | 0.97 | 0.03 |
| 9 | 1154.68 | 1629.08 | 41.08 | 0.99 | 0.01 |
| 10 | 1219.74 | 1742.04 | 42.82 | 1.00 | 0.00 |
| RMSE | | | 41.23 | | 0.0126 |
| CV | | | | 0.0107 | |

**Table 3.** The lower and upper boundaries of updating parameters for the thin plate.

| Area | Boundary | Non-Dimensional Ratio of Density | Non-Dimensional Ratio of Modulus of Elasticity | Non-Dimensional Ratio of Poisson's Ratio |
|---|---|---|---|---|
| Main | Lower | 0.1 | 0.1 | 0.1 |
| | Upper | 1.5 | 1.5 | 1.2 |
| Hole | Lower | 0.1 | 0.1 | 0.1 |
| | Upper | 1.5 | 1.5 | 1.3 |

The pre-analysis for the MUUM method was performed based on the updating parameters generated by central composite design [36]. As illustrated in Figure 4a, the swapping between the third and the fourth mode occurs frequently. The same goes for the sixth and the seventh modes, as shown in Figure 4b. Thus, four uncorrelated mode pairs are specified for the MUUM method: (A4, E3), (A3, E4), (A7, E6), and (A6, E7).

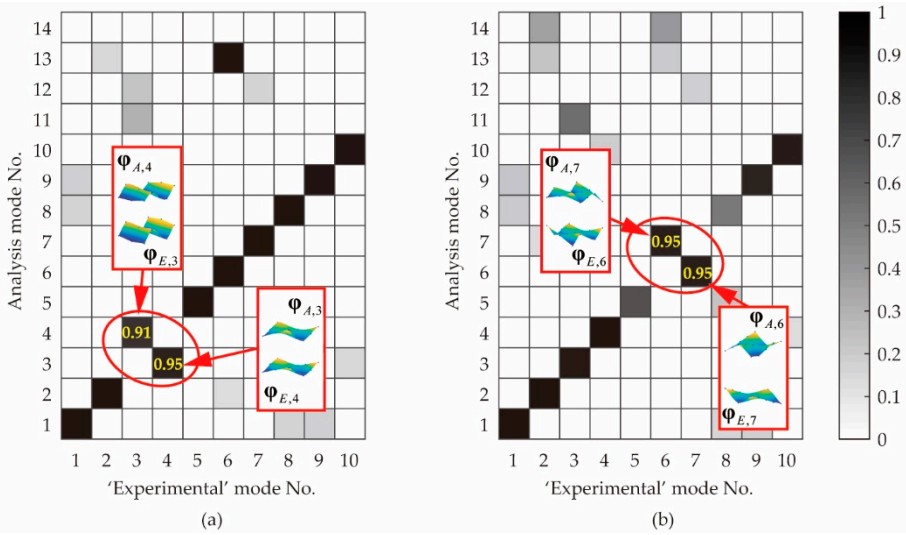

**Figure 4.** Uncorrelated mode pairs for the MUUM method based on the simulated "experimental" data. (**a**) Swapping between the third and fourth modes. (**b**) Swapping between the sixth and seventh modes.

The MUUM and CMPES method were employed to update the initial FE model, and the same optimized parameters are listed in Table 4.

**Table 4.** Optimize parameters of evolutionary algorithm.

| Parameter Name | Value | Parameter Name | Value |
|---|---|---|---|
| Number of individuals | 50 | Migration direction | forward |
| Number of elite | 5 | Migration Interval | 15 |
| Crossover probability | 0.75 | Migration probability | 0.1 |
| Maximum number of generations | 200 | Function Tolerance | 10-6 |

The updated natural frequencies and MAC values are shown in Table 5. Compared with Table 2, the updated MAC values were closer to 1. The FRE values and $RMSE_{FRE}$ of the updated models were within the range of ±1%. Therefore, both the updating methods worked well.

**Table 5.** Comparison of the updated results by the MUUM and CMPES methods based on the simulated "experimental" data.

| Mode No. | MUUM | | | | CMPES | | | |
|---|---|---|---|---|---|---|---|---|
| | Frequency (Hz) | FRE (%) | MAC | 1-MAC | Frequency (Hz) | FRE (%) | MAC | 1-MAC |
| 1 | 162.45 | 0.82 | 1.00 | 0.00 | 161.13 | 0.01 | 1.00 | 0.00 |
| 2 | 211.12 | 0.75 | 1.00 | 0.00 | 209.55 | −0.01 | 1.00 | 0.00 |
| 3 | 449.25 | 0.79 | 1.00 | 0.00 | 445.72 | 0.00 | 1.00 | 0.00 |
| 4 | 458.70 | 0.77 | 1.00 | 0.00 | 455.08 | −0.02 | 1.00 | 0.00 |
| 5 | 738.63 | 0.80 | 1.00 | 0.00 | 732.60 | −0.02 | 1.00 | 0.00 |
| 6 | 787.00 | 0.79 | 1.00 | 0.00 | 780.86 | 0.00 | 1.00 | 0.00 |
| 7 | 861.69 | 0.72 | 1.00 | 0.00 | 855.35 | −0.02 | 1.00 | 0.00 |
| 8 | 899.26 | 0.74 | 1.00 | 0.00 | 892.64 | 0.00 | 1.00 | 0.00 |
| 9 | 1163.85 | 0.79 | 1.00 | 0.00 | 1154.77 | 0.01 | 1.00 | 0.00 |
| 10 | 1229.49 | 0.80 | 1.00 | 0.00 | 1219.92 | 0.01 | 1.00 | 0.00 |
| *RMSE* | | 0.78 | | | | 0.01 | | |
| *CV* | | | | 0.00 | | | | 0.00 |

### 4.1.2. Updating Based on the Impact Modal Test Data

The thin plate was suspended via a flexible cord with nine acceleration sensors evenly arranged, as shown in Figure 5. Excitation of the thin plate was achieved using an impact hammer, and the roving hammer method was employed to excite each of the fifteen response points five times. The translational acceleration responses in the thickness direction were measured at each point to obtain the smooth frequency response function data.

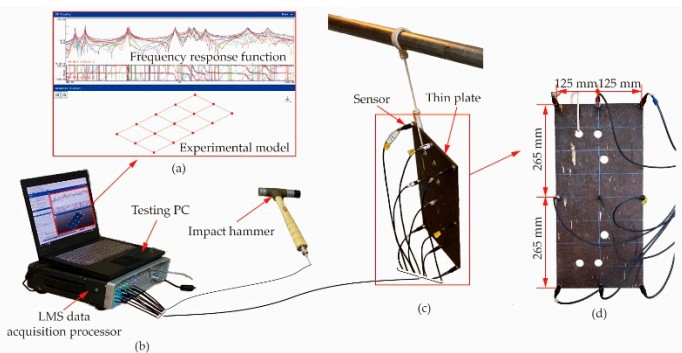

**Figure 5.** Overview of the impact modal test of the thin plate. (**a**) Screenshot of the frequency response function and experimental model. (**b**) Impact modal test instrument. (**c**) Suspension of the thin plate. (**d**) Location of the sensors.

The analytical bandwidth ranged from 100 to 1200 Hz. The stability tolerances of vibration vector, frequency, and damping were 2%, 1%, and 5%, respectively, and the model size was 100. Then, a clear stabilization diagram was obtained based on the poly-reference least squares complex exponential method [37], as shown in Figure 6. The first ten modes are marked by the red letter "s" and violaceous line in Figure 6.

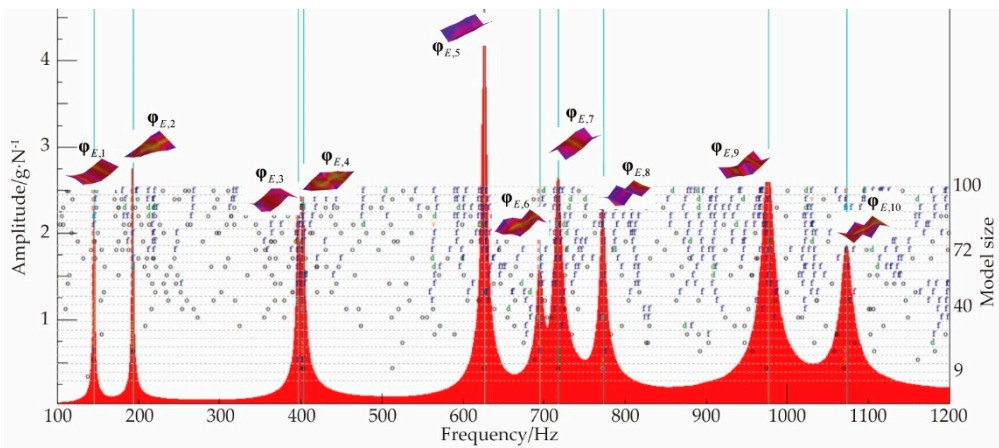

**Figure 6.** Stabilization diagram and the mode shapes of the thin plate in the impact modal test.

The natural frequencies and corresponding modal damping ratios are detailed in Table 6. Compared with Table 2, the FRE values were larger, as were the $RMSE_{FRE}$ and $CV$.

**Table 6.** Comparison between the initial analytical and impact modal results.

| Mode No. | Experimental Frequency (Hz) | Modal Damping Ratios (%) | Initial Analytical Modal Analysis | | | |
|---|---|---|---|---|---|---|
| | | | Frequency (Hz) | FRE (%) | MAC | 1-MAC |
| 1 | 145.56 | 0.34 | 227.18 | 56.08 | 0.9549 | 0.0451 |
| 2 | 193.65 | 0.24 | 303.10 | 56.52 | 0.9728 | 0.0272 |
| 3 | 397.43 | 0.30 | 624.76 | 57.20 | 0.7745 | 0.2255 |
| 4 | 403.57 | 0.40 | 658.21 | 63.10 | 0.9286 | 0.0714 |
| 5 | 627.03 | 0.32 | 1016.49 | 62.11 | 0.9918 | 0.0082 |
| 6 | 695.33 | 0.50 | 1124.19 | 61.68 | 0.7069 | 0.2931 |
| 7 | 718.01 | 0.55 | 1175.48 | 63.71 | 0.9631 | 0.0369 |
| 8 | 773.24 | 0.41 | 1222.67 | 58.12 | 0.6200 | 0.3800 |
| 9 | 976.84 | 0.62 | 1629.08 | 66.77 | 0.9492 | 0.0508 |
| 10 | 1073.56 | 0.54 | 1742.04 | 62.27 | 0.6504 | 0.3496 |
| RMSE | | | | 60.85 | | 0.2038 |
| CV | | | | | 0.1724 | |

Unlike the simulated "experimental" data, the modes obtained in the impact modal test were complex. Since all the modal damping ratios in Table 6 were less than 0.7%, the realization method was used by multiplying the modulus of each element of the complex mode shape vector by the sign of the cosine of its phase angle [10]. As shown in Figure 6, the first ten real mode shapes were obtained. Then, the following updating process was the same as Section 4.1.1.

After pre-analysis, two uncorrelated mode pairs were specified for the MUUM method: (A7, E6) and (A6, E7). Then, both the MUUM and CMPES methods were used to update the initial FE model based on the experimental modal results.

As listed in Table 7, the updated results in terms of $RMSE_{FRE}$ were the same with the value of 2.55% in both methods. For the thin plate, the discrepancies of MAC values between the FE and experimental results were affected by factors such as noncoincidence of impact and response points and added mass of acceleration sensors. The $RSME_{MAC}$ of the updated model obtained by the CMPES method was only a little smaller (from 0.2038 to 0.1902) than that of the initial analysis, but it was better than the one (0.2082) obtained by the MUUM method. The $CV$ of the updated model obtained by the CMPES method was also less than the MUUM method, which was 0.1552 and 0.1784, respectively.

**Table 7.** Comparison of the updated results by the MUUM and CMPES methods based on the impact modal test data.

| Mode No. | MUUM | | | | CMPES | | | |
|---|---|---|---|---|---|---|---|---|
| | Frequency (Hz) | FRE (%) | MAC | 1-MAC | Frequency (Hz) | FRE (%) | MAC | 1-MAC |
| 1 | 141.47 | −2.81 | 0.9489 | 0.0511 | 145.72 | 0.11 | 0.9537 | 0.0463 |
| 2 | 195.02 | 0.70 | 0.9725 | 0.0275 | 183.65 | −5.16 | 0.9716 | 0.0284 |
| 3 | 383.20 | −3.58 | 0.7507 | 0.2493 | 393.05 | −1.10 | 0.7951 | 0.2049 |
| 4 | 413.39 | 2.43 | 0.9301 | 0.0699 | 396.65 | −1.71 | 0.9276 | 0.0724 |
| 5 | 614.27 | −2.04 | 0.9917 | 0.0083 | 632.10 | 0.81 | 0.9842 | 0.0158 |
| 6 | 702.44 | 1.02 | 0.7191 | 0.2809 | 692.68 | −0.38 | 0.7244 | 0.2756 |
| 7 | 722.52 | 0.63 | 0.9723 | 0.0277 | 719.50 | 0.21 | 0.9562 | 0.0438 |
| 8 | 746.54 | −3.45 | 0.6106 | 0.3894 | 766.81 | −0.83 | 0.6559 | 0.3441 |
| 9 | 1019.28 | 4.34 | 0.9588 | 0.0412 | 1029.38 | 5.38 | 0.9437 | 0.0563 |
| 10 | 1087.17 | 1.27 | 0.6394 | 0.3606 | 1094.62 | 1.96 | 0.6657 | 0.3343 |
| RMSE | | 2.55 | | 0.2082 | | 2.55 | | 0.1902 |
| CV | | | 0.1784 | | | | 0.1552 | |

As mentioned in Section 2, the updated results are usually susceptible to the CMPs based on existing methods. For the MUUM method, since the constant groups of correlated and uncorrelated mode pairs are specified by pre-analysis, noticeable differences among updated models may occur by different groups.

To compare the robustness of the two methods, three different groups of correlated and uncorrelated mode pairs for the MUUM method were selected from full-factorial design

results, as listed in Table 8. The initial FE model was updated by the MUUM method and the CMPES method with three random initial populations, respectively.

**Table 8.** Groups of correlated and uncorrelated mode pairs for the MUUM method based on the impact modal test data.

| Group Name | Correlated Mode Pairs | Uncorrelated Mode Pairs |
| --- | --- | --- |
| MUUM | (A1, E1) (A1, E2) (A3, E3) (A4, E4) (A5, E5) (A6, E6) (A7, E7) (A8, E8) (A9, E9) (A10, E10) | (A7, E6) (A6, E7) |
| MUUM-I | (A1, E1) (A1, E2) (A3, E3) (A4, E4) (A5, E5) (A6, E6) (A7, E7) (A8, E8) (A9, E9) (A11, E10) | (A7, E6) (A8, E6) (A6, E7) (A8, E7) (A6, E8) (A7, E8) |
| MUUM-II | (A1, E1) (A1, E2) (A3, E3) (A4, E4) (A5, E5) (A6, E6) (A7, E7) (A8, E8) (A9, E9) | (A7, E6) (A8, E6) (A6, E7) (A8, E7) (A6, E8) (A7, E8) |

The $RMSE_{FRE}$, $RMSE_{MAC}$, and $CV$ of the updated model obtained by the two methods are listed in Table 9. All the values of the updated model obtained by the CMPES method were better than or equal to those obtained by the MUUM method. The $RMSE_{FRE}$ obtained by the MUUM method fluctuated wildly from 2.55% to 7.46%, compared with almost constant $RMSE_{FRE}$ obtained by the CMPES method. Similarly, the fluctuations of the $RMSE_{MAC}$ and $CV$ obtained by the MUUM method were greater than the CMPES method. Therefore, the robustness of the CMPES method is much better than the MUUM method.

**Table 9.** Robustness comparison of the MUUM and CMPES methods based on the impact modal test data.

| Group Name | $RMSE_{FRE}$ (%) | $RMSE_{MAC}$ | $CV$ | Group Name | $RMSE_{FRE}$ (%) | $RMSE_{MAC}$ | $CV$ |
| --- | --- | --- | --- | --- | --- | --- | --- |
| MUUM | 2.55 | 0.2082 | 0.1784 | CMPES | 2.55 | 0.1902 | 0.1552 |
| MUUM-I | 6.36 | 0.2147 | 0.1852 | CMPES-I | 2.55 | 0.1902 | 0.1552 |
| MUUM-II | 7.46 | 0.2161 | 0.1870 | CMPES-II | 2.54 | 0.1903 | 0.1552 |

In this example, the FE model updating of the thin plate was carried out by the MUUM and CMPES methods. The comparisons demonstrate that both the updating methods can work well. The accuracy, effectiveness, and robustness of the proposed CMPES method are better than the MUUM method.

### 4.2. Supplementary Example: Dynamic Model Updating of the F-Shaped Structure

As described in Ref. [23], a 25-frame-FE model of the F-shaped structure was updated by the MUUM and CMPES methods based on the simulated "experimental" data generated by introducing certain flexibility at the three joints and the density of the material. The updating parameters and correlated and uncorrelated mode pairings for the MUUM method were the same as Ref. [23]. According to the same optimized parameters listed in Table 4, the updated results by both methods were listed in Table 10.

Similar to Section 4.1.1, although both the updating methods work well, the $RMSE_{FRE}$ obtained by the CMPES method is still smaller than the MUUM method.

**Table 10.** Comparison of the updated results of the F-shaped structure by the MUUM and CMPES methods based on the simulated "experimental" data.

| Mode No. | "Experimental" Frequency (Hz) | MUUM | | | | CMPES | | | |
|---|---|---|---|---|---|---|---|---|---|
| | | Frequency (Hz) | FRE (%) | MAC | 1-MAC | Frequency (Hz) | FRE (%) | MAC | 1-MAC |
| 1 | 12.01 | 12.17 | 1.34 | 1.00 | 0.00 | 12.01 | 0.00 | 1.00 | 0.00 |
| 2 | 54.73 | 54.86 | 0.24 | 1.00 | 0.00 | 54.73 | 0.00 | 1.00 | 0.00 |
| 3 | 62.38 | 62.52 | 0.21 | 1.00 | 0.00 | 62.38 | 0.00 | 1.00 | 0.00 |
| 4 | 260.71 | 260.85 | 0.05 | 1.00 | 0.00 | 260.71 | 0.00 | 1.00 | 0.00 |
| 5 | 1110.40 | 1110.46 | 0.01 | 1.00 | 0.00 | 1110.41 | 0.00 | 1.00 | 0.00 |
| 6 | 1150.50 | 1150.57 | 0.01 | 1.00 | 0.00 | 1150.52 | 0.00 | 1.00 | 0.00 |
| 7 | 1208.40 | 1208.67 | 0.02 | 1.00 | 0.00 | 1208.37 | 0.00 | 1.00 | 0.00 |
| 8 | 2130.40 | 2130.63 | 0.01 | 1.00 | 0.00 | 2130.46 | 0.00 | 1.00 | 0.00 |
| 9 | 2245.80 | 2245.80 | 0.00 | 1.00 | 0.00 | 2245.77 | 0.00 | 1.00 | 0.00 |
| *RMSE* | | | 0.46 | | | | 0.00 | | |
| *CV* | | | | 0.00 | | | | | 0.00 |

### 4.3. Engineering Example: Dynamic Model Updating of an Intermediate Case

This section describes the updating of an intermediate case (IMC) of a gas turbine to validate the potential of the proposed CMPES method for complex engineering structures. IMC is a vitally important transition channel between a low-pressure compressor and a high-pressure compressor, and it is the main load-bearing component because the thrust generated by the engine is transmitted through it. The IMC is welded by titanium alloy consisting of the outer casing, strut, splitter, and inner casing. The cross-sectional shapes of the twelve hollow struts (abbreviated as 1#–12#) are not the same due to the different functions. Holes with different apertures, cross-sectional shapes, and locations are processed on the multi-layer thin-walled cases. In addition, many installation accessories are assembled on it, including mount, rectifier rotating rocker arms, and lubricating oil ducts. Therefore, it is a great challenge to establish an accurate FE model.

The initial FE model of the IMC, ignoring local details such as holes, chamfers, and accessories, was built by 18,936 three-dimensional elements, as illustrated in Figure 7. According to the cross-sectional shapes, the struts can be divided into three kinds: wide strut (1#), middle struts (3#, 7#, and 11#), and narrow struts (2#, 4#, 5#, 6#, 8#, 9#, 10#, and 12#). To simplify the dynamic FE model, the irregular cross-sectional shapes of these struts were all equivalent to a hollow thin-walled rectangle with constant bending stiffness of the narrow struts, shown in Figure 7b. All the elements were assigned to the same material attribute of titanium alloy: density was 4500 kg/m$^3$, Young's modulus was 109.8 GPa, and Poisson's ratio was 0.3.

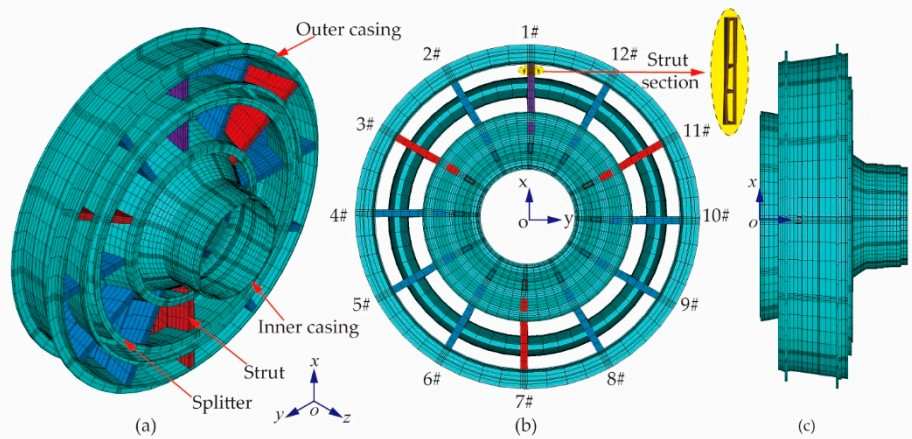

**Figure 7.** The initial dynamic FE model of the IMC. (**a**) Isometric view. (**b**) Left view and partial view of the strut section. (**c**) Front view.

In order to update the FE model, the multiple reference impact technique was employed to conduct the quasi-free–free experimental modal test in the radial plane (the *xoy* plane in Figure 7). As shown in Figure 8, the testing IMC was suspended horizontally by a soft rope reeved through bolt holes on the rear flange of the outer ring to approximately represent the free–free boundary in the radial plane. Considering the distribution characteristics of the struts and the measurable location of response and excitation points, the test model of 372 response points for the testing IMC was established. Only twelve translational acceleration sensors were bolted to the flanges, marked with arrows in the upper left corner of Figure 8. All the response points were excited five times by an impact hammer one by one to measure the response of each sensor. The analytical bandwidth ranged from 100 to 880 Hz, about four times the operating speed. Then, the first fourteen incomplete modes were identified in the range based on the poly-reference least squares complex exponential method.

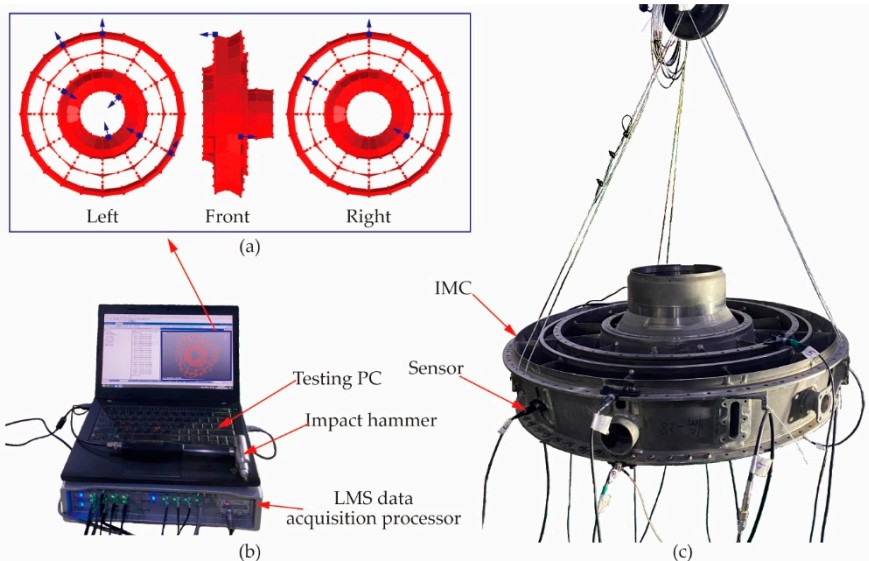

**Figure 8.** Physical photo of free–free experimental modal test of the IMC. (**a**) Experimental model and the location of the sensors. (**b**) Impact modal test instrument. (**c**) Suspension of the IMC.

The modal analysis with free–free boundary condition was performed on the initial FE model. The first twenty natural frequencies and mode shapes were determined. After accomplishing the coordinate pairing in the response direction between the FE nodes and the experimental response points (see Figure 8), the MAC values in the radial plane between the initial analytical and incomplete experimental modes (realization method is the same as Section 4.1.2) were calculated.

Figure 9 shows the correlation analysis results of the initial analytical and experimental modes. It can be seen that the maximal MAC value was 0.75, which was much smaller than that in Table 6. When performing correlation pairing with a larger MAC threshold, such as 0.5, the case 0↔n occurs frequently. If a smaller MAC threshold of 0.25 is used, though the number of pairing modes increases, the case 1↔n increases sharply. Therefore, the correlation pairing results change significantly due to different MAC thresholds.

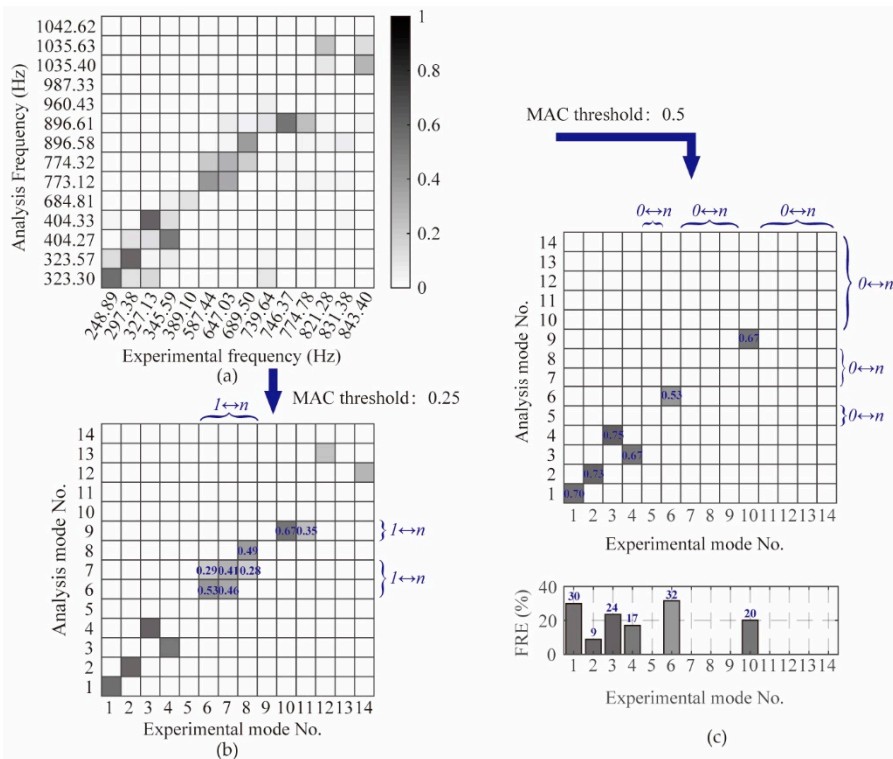

**Figure 9.** Correlation analysis results of the initial analytical and experimental modes of the IMC. (**a**) MAC matrix. (**b**) Correlation analysis results described by MAC matrix with the MAC threshold of 0.25. (**c**) Correlation analysis results described by MAC matrix and FRE bar graph with the MAC threshold of 0.5.

When the MAC threshold of 0.5 was used, the initial analytical results could only be paired with six experimental modes. The order of the third and fourth modes was swapped. The majority of FRE values of these six modes were also more than 20%. Moreover, the analytical natural frequencies were higher than the corresponding experimental results. The $RMSE_{FRE}$ and $RMSE_{MAC}$ of the initial analysis results were 22.68% and 0.3709, respectively. The discrepancies between them were apparent. Therefore, the initial FE model needed to be updated.

Considering the structural characteristics of the IMC and the simplifications of the dynamic FE model, the density and Young's modulus of the four groups of components were used as updating parameters distinguished by colors in Figure 7: the outer casing, splitter, and inner casing in green, the wide strut in purple, the middle struts in red, and the narrow struts in blue. Similar to Section 4.1, the non-dimensional ratios of material parameters to the corresponding titanium alloy values were used. The lower and upper boundaries of all updating parameters were 0.50 and 1.50. The optimized parameters used in this example are the same listed in Table 4. Then, the initial FE model was updated by the proposed CMPES method. More details of the varied correlation pairing results due to individual differences in the same generation and evolution were given to show the advantage of the correlated mode auto-pairing strategy and population evolution screening mechanism.

The objective function values $\mathbf{J}^{(0)}$ of all individuals in the initial population were calculated by Equation (18), shown in Figure 10a. Three typical individuals in the initial population were marked: the worst individual (individual no. 32, box), the individual closest to the mean (individual no. 34, diamond), and the best individual (individual no. 8, circle). The MAC matrices and corresponding FRE bar graphs of these three individuals determined by the correlated mode auto-pairing strategy are shown in Figure 10b–g. The numbers of correlation pairing modes and the swapped mode orders differed. The corresponding FRE values changed significantly among individuals. It is not possible

to capture these differences by using a constant correlation pairing method based on pre-analysis.

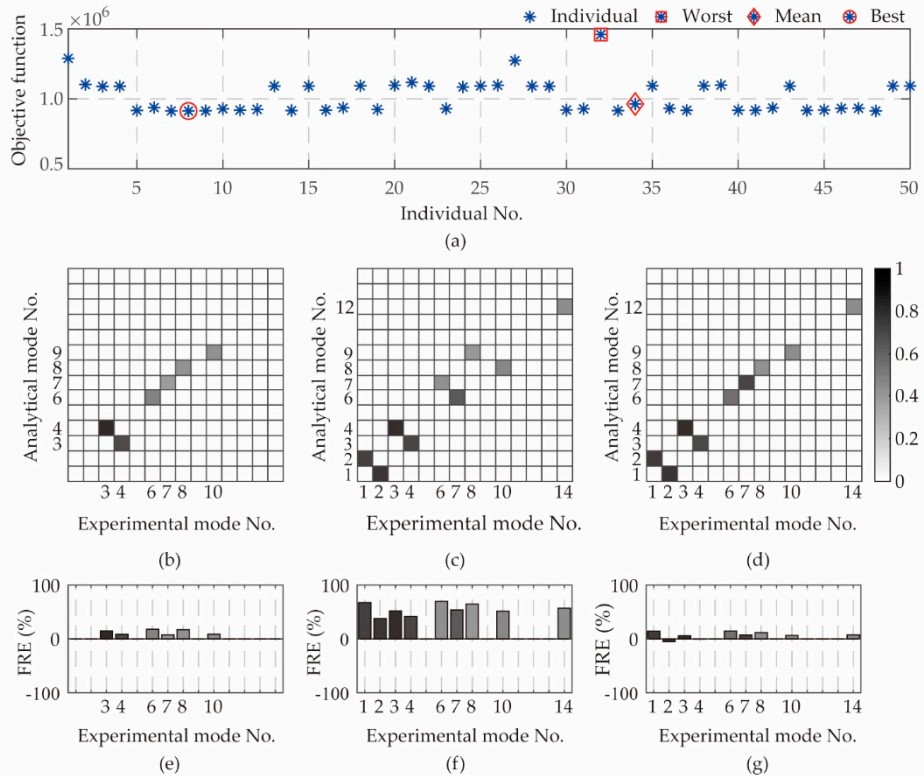

**Figure 10.** Objective function values and correlation pairing differences in the initial population. (**a**) Objective function values of $\mathbf{J}^{(0)}$; (**b**) MAC matrix of $\mathbf{MAC}^{(0,32)}$; (**c**) MAC matrix of $\mathbf{MAC}^{(0,34)}$; (**d**) MAC matrix of $\mathbf{MAC}^{(0,8)}$; (**e**) FRE bar graphs of $\mathbf{FRE}^{(0,32)}$; (**f**) FRE bar graphs of $\mathbf{FRE}^{(0,34)}$; (**g**) FRE bar graphs of $\mathbf{FRE}^{(0,8)}$.

The objective function values of the best individual in each population are shown in Figure 11. The value decreasing with evolution indicated that the correlation pairing results were improved. Convergence was achieved in the 83rd generation, for the average change in objective function value in the last-fifteen-generation population was less than the given function tolerance in Table 4.

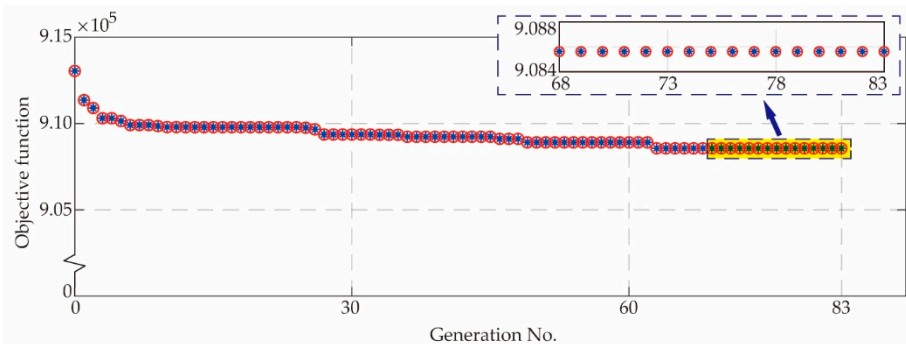

**Figure 11.** Objective function value of the best individual changes with the evolution.

The objective function values and correlation pairing differences among the typical individuals in the last-generation population are shown in Figure 12. Compared with Figure 10, the discrete degree of objective function values was significantly reduced. The numbers of correlation pairing modes of the typical individuals were increased. The mode swapping was improved. The FRE values were also significantly reduced.

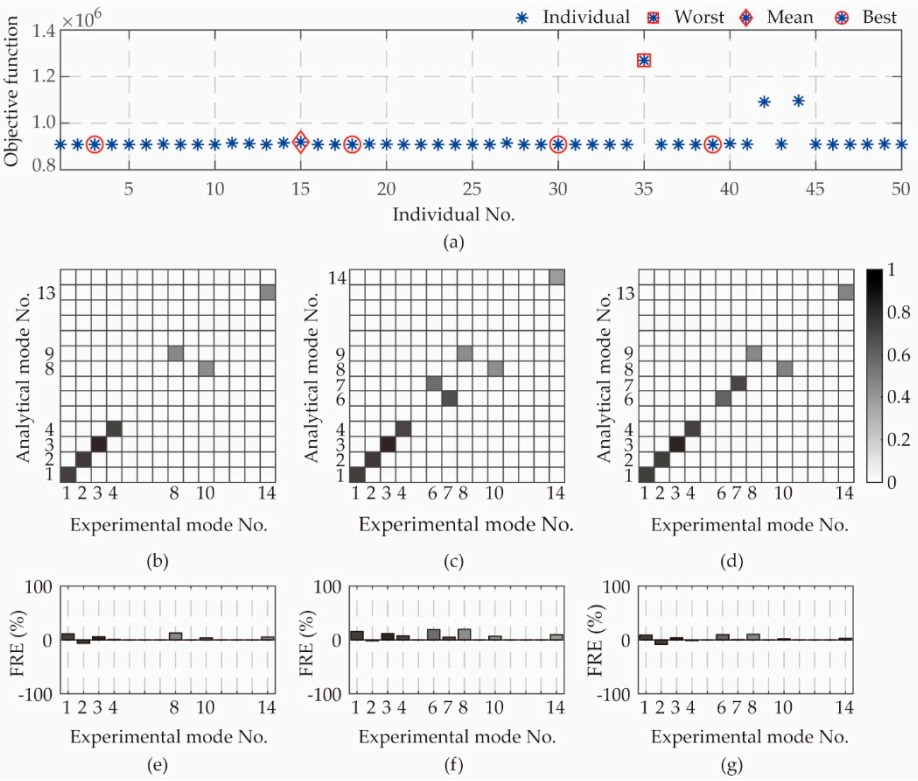

**Figure 12.** Objective function values and correlation pairing differences in the last-generation population. (**a**) Objective function values of $\mathbf{J}^{(83)}$; (**b**) MAC matrix of $\mathbf{MAC}^{(83,35)}$; (**c**) MAC matrix of $\mathbf{MAC}^{(83,15)}$; (**d**) MAC matrix of $\mathbf{MAC}^{(83,3)}$; (**e**) FRE bar graphs of $\mathbf{FRE}^{(83,35)}$; (**f**) FRE bar graphs of $\mathbf{FRE}^{(83,15)}$; (**g**) FRE bar graphs of $\mathbf{FRE}^{(83,3)}$.

The correction pairing results of the best individual in the last-generation population are listed in Table 11. Compared with the best individual in the initial population (see Figure 10d,g), the number of correlation pairing modes was increased from six to nine after updating, which improved the exploitation of experimental modal data. The MAC values of the corresponding modes were increased. The $RMSE_{MAC}$ was decreased to 0.2591. Besides, the FRE values of the other modes were all within 10% except the eighth mode. The $RMSE_{FRE}$ was reduced to 6.50%.

**Table 11.** The final updated modal results for the IMC.

| Mode No. | Mode Description | Experimental Frequency (Hz) | Updated Analytical Frequency (Hz) | FRE (%) | MAC | 1-MAC |
|---|---|---|---|---|---|---|
| 1 | Radial | 248.89 | 270.87 | 8.83 | 0.8752 | 0.1248 |
| 2 | Radial | 297.38 | 272.17 | −8.48 | 0.8752 | 0.1248 |
| 3 | Radial | 327.13 | 339.00 | 3.63 | 0.9420 | 0.0580 |
| 4 | Radial | 345.59 | 341.18 | −1.28 | 0.8616 | 0.1384 |
| 5 | Torsional | 389.10 | | | | |
| 6 | Radial | 587.44 | 644.32 | 9.68 | 0.7409 | 0.2591 |
| 7 | Radial | 647.03 | 649.33 | 0.36 | 0.8444 | 0.1556 |
| 8 | Radial | 689.50 | 761.62 | 10.46 | 0.5987 | 0.4013 |
| 9 | Axial | 739.64 | | | | |
| 10 | Radial | 746.37 | 760.85 | 1.94 | 0.6187 | 0.3813 |
| 11 | Axial | 774.78 | | | | |
| 12 | Axial | 821.28 | | | | |
| 13 | Local | 831.38 | | | | |
| 14 | Radial | 843.40 | 867.74 | 2.89 | 0.6091 | 0.3909 |
| *RMSE* | | | | 6.50 | | 0.2591 |
| *CV* | | | | | 0.1736 | |

By further comparison with the initial analytical results in Figure 9, the discrepancies of the final updated results were noticeably decreased. Only for these six modes (first, second, third, fourth, sixth, and tenth) paired before and after updating were most of the FRE values reduced from more than 20% to less than 10%. The MAC values of the corresponding modes were increased. The mode swapping was improved to a certain extent. Furthermore, the $RMSE_{FRE}$ was significantly reduced from 22.68% to 6.12%. The $RSME_{MAC}$ was reduced from 0.3709 to 0.2037. The $CV$ was reduced from 0.1802 to 0.1605.

Based on the proposed CMPES method, all the radial modes (excluding the local mode) within the range of four times of operating speed were auto screened out and updated well. The comparisons between the updated analytical and experimental mode shapes for these paired modes are shown in Figure 13.

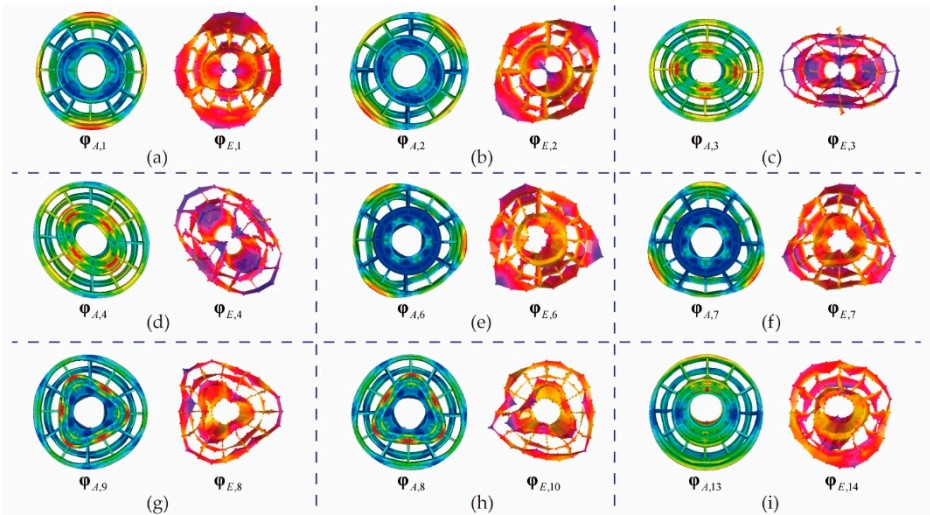

**Figure 13.** Comparison between the updated analytical and experimental mode shapes. (**a**) The 1st analytical and the 1st experimental mode shapes; (**b**) the 2nd analytical and the 2nd experimental mode shapes; (**c**) the 3rd analytical and the 3rd experimental mode shapes; (**d**) the 4th analytical and the 4th experimental mode shapes; (**e**) the 6th analytical and the 6th experimental mode shapes; (**f**) the 7th analytical and the 7th experimental mode shapes; (**g**) the 9th analytical and the 8th experimental mode shapes; (**h**) the 8th analytical and the 10th experimental mode shapes; (**i**) the 13th analytical and the 14th experimental mode shapes.

In this example, even though the discrepancies of the updated dynamic FE and experimental model were not absolutely eliminated, the improvement in updating results demonstrates the great potential of the proposed method for complex engineering problems.

## 5. Conclusions

The CMPES method is proposed to deal with the difficulties in pairing inaccurate analytical and incomplete experimental modal data when updating the dynamic FE model of complex engineering structures. The conclusions are as follows:

(1) To solve the problem that modal data cannot be fully exploited when natural frequencies and mode shapes of complex structures change dramatically, the correlated mode auto-pairing strategy is proposed. The one-to-one CMPs are determined adaptively by the auto-pairing strategy before each iteration based on the one-to-one symbiotic relationship between natural frequencies and mode shapes. This strategy, liberating from dependence on demanding pre-analysis, is suitable for all the 1↔1, 1↔n, and 0↔n cases in dynamic FE model updating.

(2) To further screen the CMPs determined by the auto-pairing strategy in each generation and search the global minimum, the population evolution mechanism is used to

simultaneously urge the updating parameters and the CMPs towards the ideal results. Combined with the auto-pairing strategy, the adaptive switch from penalized to correlated subitem can screen the CMPs further during iteration to ensure that all the potential of analytical or experimental modes can be fully exploited.

(3) The examples of a thin plate with non-uniform thickness and small holes validated the accuracy and effectiveness of the proposed method. In the updating of the IMC with different cross-sectional shapes of hollow struts and multi-layer thin-walled complex structure, all the radial modes within the range of four times the operating speed were auto-screened out and updated well. The updated results show the great potential of the proposed CMPES method for complex engineering problems.

**Author Contributions:** Conceptualization, Y.Z. and Z.J.; methodology, H.S.; software, H.S.; validation, Y.Z. and H.S.; formal analysis, H.S.; investigation, H.S.; resources, Y.Z. and Z.J.; data curation, H.S.; writing—original draft preparation, H.S.; writing—review and editing, Y.Z. and Z.J. All authors have read and agreed to the published version of the manuscript.

**Funding:** This research was funded by the Young Scientists Fund of the National Natural Science Foundation of China, grant number 51905025.

**Institutional Review Board Statement:** Not applicable.

**Informed Consent Statement:** Not applicable.

**Data Availability Statement:** Not applicable.

**Conflicts of Interest:** The authors declare no conflict of interest.

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
