# Peer review of "Dynamic Finite Element Model Updating Based on Correlated Mode Auto-Pairing and Adaptive Evolution Screening"

_applsci, doi:10.3390/app12063175_

Round 1
Reviewer 1 Report
Authors have studied A method for dynamic finite element (FE) model updating based on correlated mode auto-pairing and adaptive evolution screening (CMPES) is proposed to overcome the difficulties in pairing inaccurate analytical modal data and incomplete experimental modal data. In each generation, the correlated mode pairings (CMPs) are determined by Modal Assurance Criterion (MAC) values and the symbiotic natural frequency errors, according to an auto-pairing strategy. The objective function values constructed by correlated and penalized subitems are calculated to screen the better individuals. Then both the updating parameters and the CMPs can be adjusted adaptively to simultaneously approach the ideal results during the iteration of population evolution screening. Two examples, a thin plate with small holes and an intermediate case with multi-layer thin-walled complex structure, were presented to validate the accuracy, effectiveness, and engineering application potential of the proposed method. I recommend this article for publication if authors are adequately address the following comments.
1) What is the motivation of the present study. Also, provide the physical applications of the considered problem.
2) Authors have to validate their results with two are more published experimental works.
3) Authors has to recheck their manuscript for typos.
4) Add the following recent articles to enhance quality of manuscript.
Finite Element Analysis of Thermo-Diffusion and Diffusion-Thermo Effects on Convective Heat and Mass Transfer Flow Through a Porous Medium in Cylindrical Annulus.
Reviewer 2 Report
Solutions to overcome the difficulties in pairing inaccurate analytical modal data and incomplete experimental modal data are always of scientific interest. Presented article is giving additional possibility to solve this problem by a method for dynamic finite element (FE) model updating.
I would recommend the following improvements:
1. To be added a new figure, following Figure 6 with a detailed drawing of the thin plate with the dimensions and location of the holes and the location of the sensors.
2. There is no comparison with similar studies in the analysis of results. Would be recommended to be included.
Reviewer 3 Report
Dear Authors,
I have read your research paper carefully.
I have found quite enjoyable to read it. Nevertheless, I have found a few issues along the document that you need to properly address. Attached is a suggestion list that you need to review, please feel free to use it at your convenience in order to clarify.
Another set of suggestions are listed below:
1.- Please make a deep English review of the document.
2.- Improve the quality of whole figures as well as the caption. Remember that figures must stand by its own.
3.- Can you please specify the computational time whilst using a normal method and yours?
